# Rain Streak Removal for Single Images Using Conditional Generative Adversarial Networks

**Prasad Hettiarachchi** [1,2], **Rashmika Nawaratne** [1,*], **Damminda Alahakoon** [1], **Daswin De Silva** [1] **and Naveen Chilamkurti** [3]

1 Research Centre for Data Analytics and Cognition, La Trobe University, Melbourne, VIC 3086, Australia; prasad.12@cse.mrt.ac.lk (P.H.); D.Alahakoon@latrobe.edu.au (D.A.); D.DeSilva@latrobe.edu.au (D.D.S.)
2 Department of Computer Science and Engineering, University of Moratuwa, Moratuwa 10400, Sri Lanka
3 Department of Computer Science and Information Technology, La Trobe University, Melbourne, VIC 3086, Australia; N.Chilamkurti@latrobe.edu.au
* Correspondence: B.Nawaratne@latrobe.edu.au

**Abstract:** Rapid developments in urbanization and smart city environments have accelerated the need to deliver safe, sustainable, and effective resource utilization and service provision and have thereby enhanced the need for intelligent, real-time video surveillance. Recent advances in machine learning and deep learning have the capability to detect and localize salient objects in surveillance video streams; however, several practical issues remain unaddressed, such as diverse weather conditions, recording conditions, and motion blur. In this context, image de-raining is an important issue that has been investigated extensively in recent years to provide accurate and quality surveillance in the smart city domain. Existing deep convolutional neural networks have obtained great success in image translation and other computer vision tasks; however, image de-raining is ill posed and has not been addressed in real-time, intelligent video surveillance systems. In this work, we propose to utilize the generative capabilities of recently introduced conditional generative adversarial networks (cGANs) as an image de-raining approach. We utilize the adversarial loss in GANs that provides an additional component to the loss function, which in turn regulates the final output and helps to yield better results. Experiments on both real and synthetic data show that the proposed method outperforms most of the existing state-of-the-art models in terms of quantitative evaluations and visual appearance.

**Keywords:** deep learning; generative adversarial networks; traffic surveillance image processing; image de-raining

## 1. Introduction

Rain is a common weather condition that negatively impacts computer vision systems. Raindrops appear as bright streaks in images due to their high velocity and light scattering. Since image recognition and detection algorithms are designed for clean inputs, it is essential to develop an effective mechanism for rain streak removal.

A number of research efforts have been reported in the literature focusing on restoring rain images, and different approaches have been taken. Some have attempted to remove rain streaks using video [1–3], while other researchers have focused on rain image recovery from a single image by considering the image as a signal separation task [4–6].

Since rain streaks overlap with background texture patterns, it is quite a challenging task to remove the rain streaks while maintaining the original texture in the background. Most of the times, this results in over-smoothed regions that are visible in the background after the de-raining process. De-raining algorithms [7,8] tend to over de-rain or under de-rain the original image. A key limitation in the traditional, handcrafted methods is that the feature learning is manual and designed to deal only with certain types of rain streaks, and they do not perform well with varying scales, shapes, orientations, and densities

of raindrops [9,10]. In contrast, by using convolutional neural networks (CNNs), the feature learning process becomes an integral part of the algorithm and is able to unveil many hidden features. Convolutional neural network-based methods [11–13] have gained huge improvements in image de-raining during the last few years. These methods try to figure out a nonlinear mapping between the input rainy image and the expected ground truth image.

Still, there is potential for improvements and optimizations within CNN-based image de-raining algorithms, which could lead to more visually appealing and accurate results. Instead of being just constrained to characterizing rain streaks, visual quality should also be considered when defining the optimization functions, which will result in improving the visual appeal of test results. When defining the objective function, it should consider the fact that the performance of vision algorithms, such as classification/detection, should not be affected by the presence of rain streaks. The addition of this discriminative information ensures that the output is indistinguishable from its original counterpart.

Generative modeling is an unsupervised learning task in machine learning that involves automatically discovering and learning the patterns in input data in such a way that the model can be used to generate new examples that are indistinguishable from reality. The concept of generative adversarial networks (GANs) was originally presented in [14] and has gained a high level of interest, with several successful applications and directions reported within a short period in the machine learning community. Existing CNN-based mechanisms only consider either $L_1$(Least Absolute Deviations) or $L_2$ (Least Square Errors) errors, whereas in conditional GANs, they have additional adversarial loss components, which result in very good, qualitative, visually appealing image outputs.

In our approach, we propose a conditional generative adversarial network-based framework for rain streak removal. Our model consists of a densely connected generator (G) network and a CNN-based discriminator (D) network. The generator network converts rainy images to de-rained images in such a way that it fools the discriminator network. In certain scenarios, traditional GANs tend to make output images more artificial and visually displeasing. To mitigate this issue, we have introduced a conditional CNN with skip connections for the generator. Skip connections guarantee better convergence by efficiently leveraging features from different layers of the network. The proposed model is based on the Pix2Pix framework by Isola et al. [15] and the conditional generative adversarial networks originally proposed by Fu et al. [16]. We have also used the source codes provided by authors of LPNet [17] and GMM [18] for quantitative and qualitative comparisons of the proposed model.

This paper makes the following contributions:

1.　Propose a conditional, GAN-based deep learning architecture to remove rain streaks from images by adapting U-Net architecture-based CNN for single image de-raining.
2.　Develop a classifier to identify whether the generated image is real or fake based on intra-convolutional "PatchGAN" architecture.
3.　Due to the lack of access to the ground truth of rainy images, we present a new dataset synthesizing rainy images using real-world clean images, which are used as the ground truth counterpart in this research.

The paper is organized as follows: In Section 2, we provide an overview of related methods for image de-raining and the basic concepts behind cGANs. Section 3 describes the proposed model (CGANet—Conditional Generative Adversarial Network model) in detail with its architecture. Section 4 describes the experimental details with evaluation results. Section 5 provides the conclusion. Implementation details and the dataset used for the experiments are publicly available at GitHub (https://github.com/prasadmaduranga/CGANet (accessed on 11 December 2020)).

## 2. Related Work

In the past, numerous methods and research approaches have been proposed for image de-raining. These methods can be categorized as single image-based methods and

video-based methods. With the evolution of neural networks, deep learning-based methods have become more dominant and efficient compared to past state-of-the-art methods.

### 2.1. Single Image-based Methods

Single image-based methods have limited access to information compared to video-based methods, which makes it more challenging to remove the rain streaks. Single image-based methods include low-rank approximations [3,19], dictionary learning [4,5,20], and kernel-based methods [21]. In [4], the authors decomposed the image into high- and low-frequency components and recognized the rain streaks by processing the high-frequency components. Other mechanisms have used gradients [22] and mixture models [18] to model and remove rain streaks. In [18], the authors introduced a patch-based prior for both clean and rainy layers using Gaussian mixture models (GMM). The GMM prior for rainy layers was learned from rainy images, while for the clean images, it was learned from natural images. Nonlocal mean filtering and kernel regression were used to identify rain streaks in [21].

### 2.2. Video-based Methods

With the availability of inter-frame information, video-based image de-raining is relatively more effective and easier compared to single image de-raining. Most research studies [1,23,24] have focused on detecting potential rain streaks using their physical characteristics and removing them using image restoration algorithms. In [25], the authors divided rain streaks into dense and sparse groups and removed the streaks using a matrix decomposition algorithm. Other methods have focused on de-raining in the Fourier domain [1] using Gaussian mixture models [23], matrix completions [24], and low-rank approximations [3].

### 2.3. Deep Learning based Methods

Deep learning-based methods have gained much popularity and success in a variety of high-level computer vision tasks in the recent past [26–28] as well as in image processing problems [29–31]. Deep learning was introduced for de-raining in [11] where a three-layer CNN was used for removing rain streaks and dirt spots in an image that had been taken through glass. In [12], a CNN was proposed for video-based de-raining, while a recurrent neural network was adopted by Liu in [32]. The authors in [33] proposed a residual-guide feature fusion network for single image de-raining. A pyramid of networks was proposed in [17], which used the domain-specific knowledge to reinforce the learning process.

CNNs learn to minimize a loss function, and the loss value itself decides the quality of output results. Significant design efforts and domain expertise are required to define an effective loss function. In other words, it is necessary to provide the CNN with what the user requires to minimize. Instead, if it is possible to set a high-level, general goal such as "make the output image indistinguishable from the target images", then the CNN can automatically learn a loss function to satisfy the goal. This is the basic underlying concept behind generative adversarial networks (GANs).

### 2.4. Generative Adversarial Networks

Generative adversarial networks [14] are unsupervised generative models that contain two deep neural networks. The two neural networks are named as the generator ($G$) and discriminator ($D$) and are trained parallelly during the training process. GAN training can be considered to be a two-player min-max game where the generator and discriminator compete with each other to achieve each other's targeted goal. The generator is trained to learn a mapping from a random noise vector ($z$) in latent space to an image ($x$) in a target domain: $G(z) \to x$. The discriminator (D) learns to classify a given image as a real (output close to 1) image or a fake (output close to 0) image from the generator (G): $D(x) \to$ [0.1]. Both the generator and decimator can be considered as two separate neural networks trained from backpropagation, and they have separate loss functions. Figure 1 shows

the high-level architecture of the proposed conditional GAN model. The generator will try to generate synthetic images that resemble real images to fool the discriminator. The discriminator learns how to identify the real images from the generated synthetic images from the generator.

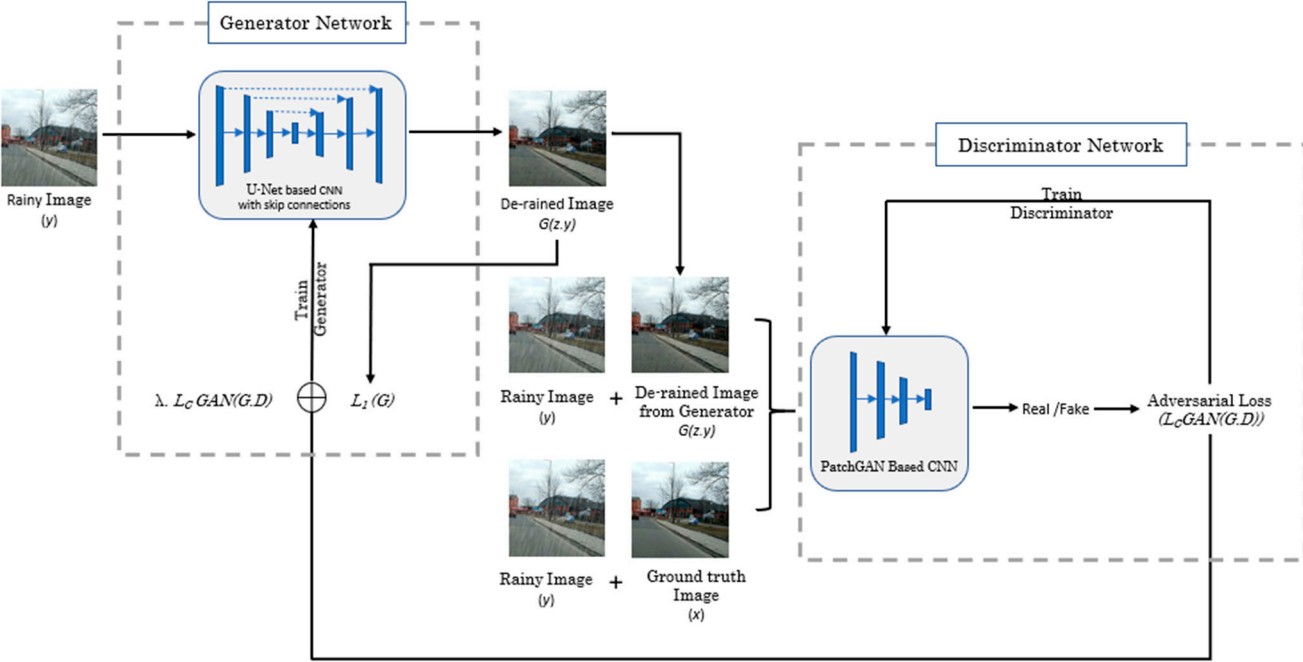

**Figure 1.** High-level architecture of the proposed model (CGANet).

The widest adaptation of GANs is for data augmentation, or that is to say, to learn from existing real-world samples and generate new samples consistent with the distribution. Generative modeling has been used in a wide range of application domains including computer vision, natural language processing, computer security, medicine, etc.

Xu et al. [34] used GANs for synthesizing image data to train and validate perception systems for autonomous vehicles. In addition to that, [35,36] used GANs for data fusion for developing image classification models while mitigating the issue of having smaller datasets. Furthermore, GANs were used for augmenting datasets for adversarial training in [37]. To increase the resolution of images, a super-resolution GAN was proposed by Ledig et al. [38], which took a low-resolution image as the input and generated a high-resolution image with $4\times$ upscaling. To convert the image content from one domain to another, an image-to-image translation approach was proposed by Isola et al. [15] using CGANs. Roy et al. [39] proposed a TriGAN, which could solve the problem of image translation by adapting multiple source domains. Experiments showed that the SeqGAN proposed in [40] outperformed the traditional methods used for music and speech generation. In the computer security domain, Hu and Tan [41] proposed a GAN-based model to generate malware. For private product customization, Hwang et al. [42] proposed GANs to manufacture medical products.

## 3. Proposed Model

The proposed approach uses image-to-image translation for the image de-raining task. In a GAN, the generator produces the output based on the latent variable or the noise variable ($z$). However, in the proposed approach, it is necessary for a correlation to exist between the source image and the generator output image. We have applied the conditional GAN [16], which is a variant of the traditional GAN that takes additional information, y, as the input. In this case, we provide a source image with rain streaks as

additional information for both the generator and discriminator. $x$ represents the target image.

The objective of a conditional GAN is as follows:

$$L_{cGAN}(G.D) = E_{x \sim pdata(X)}[log(D(x.y))] + E_{z \sim pz(Z)}[log(1 - D(G(z.y).y)) \tag{1}$$

where $pdata(X)$ denotes the real data probability distribution defined in the data space $X$, and $pz(Z)$ denotes the probability distribution of the latent variable $z$ defined on the latent space $Z$. $E_{x \sim pdata(X)}$ and $E_{z \sim pz(Z)}$ represent the expectations over the data spaces $X$ and $Z$ respectively. $G(.)$ and $D(.)$ represent the non-linear mappings of the generator and discriminator networks respectively.

In an image de-raining task, the higher-order color and texture information has to be preserved during the image translation. This has a significant impact on the visual performance of the output. Adversarial loss alone is not sufficient for this task. The loss function should be optimized so that it penalizes the perceptual differences between the output image and the target image.

Our implementation architecture is based on the work of Isola's [15] Pix2Pix framework. It learns a mapping from an input image to an output image along with the objective function to train the model. In Pix2Pix, it suggests $L_1$ (mean absolute error) loss instead of $L_2$ (mean squared error) loss for the GAN objective function, since it encourages less blurring in the generator output. $L_1$ loss averages the pixel level absolute difference between the target image and the generated image $G(z.y)$ over the image space $x.y.z$.

$$L_1(G) = Ex.y.z[\| x - G(z.y) \|] \tag{2}$$

Finally, the loss function for this work is as follows:

$$L(G.D) = L_{cGAN}(G.D) + \lambda L_1(G) \tag{3}$$

Lambda ($\lambda$) is a hyperparameter that controls the weights of the terms. In this case, we kept lambda = 100 [15]. When training the model, lambda was increased to train a discriminator and minimized to train a generator. The final objective was to identify the generator $G^*$ by solving the following optimization problem:

$$G^* = \arg min_G \ max_D (L_{cGAN}(G,D) + \lambda L_1(G)) \tag{4}$$

*Model Overview*

- Generator Network

In image-to-image translations, it is necessary to map a high-resolution input grid to a high-resolution output grid. Though the input and output images differ in appearance, they share the same underlying structure, and as such, it is necessary to consider this factor when designing the generator architecture. Most previous work used the encoder-decoder network [43] for such scenarios. In encoder-decoder CNN, the input is progressively downsampled until the bottleneck layer, where the process gets reversed and starts to upsample the input data. Convolutional layers use $4 \times 4$ filters and strides with size 2 for downsampling. The same size of kernel is used for transpose convolution operation during upsampling. Each convolution/deconvolution operation is followed by batch normalization and Rectified Linear Unit (ReLU) activation. Weights of the generator are updated depending on the adversarial loss of the discriminator and the $L_1$ loss of the generator. Architecture details are shown in Table 1.

**Table 1.** Generator architecture of the CGANet model.

| Generator Architecture |
| :---: |
| Input(256 × 256), Num_c = 3 |
| Downsampling: 4 × 4 Convolution + BN + ReLu, Output: 128 × 128, Num_c: 64 |
| Downsampling 4 × 4 Convolution + BN + ReLu, Output: 64 × 64, Num_c: 128 |
| Downsampling 4 × 4 Convolution + BN + ReLu, Output: 64 × 64, Num_c: 128 |
| Downsampling: 4 × 4 Convolution + BN + ReLu, Output: 32 × 32, Num_c: 256 |
| Downsampling: 4 × 4 Convolution + BN + ReLu, Output: 16 × 16, Num_c: 512 |
| Downsampling 4 × 4 Convolution + BN + ReLu, Output: 128 × 128, Num_c: 512 |
| Downsampling 4 × 4 Convolution + BN + ReLu, Output: 8×8, Num_c: 512 |
| Downsampling 4 × 4 Convolution + BN + ReLu, Output: 4 × 4, Num_c: 512 |
| Downsampling 4 × 4 Convolution + BN + ReLu, Output: 2 × 2, Num_c: 512 |
| Downsampling 4 × 4 Convolution + BN + ReLu, Output: 1 × 1, Num_c: 512 |
| Upsampling 4 × 4 Convolution + BN + ReLu, Output: 2 × 2, Num_c: 512 |
| Concatenation: Input (2×2×512), (2 × 2 × 512), Output (2 × 2 × 1024) |
| Upsampling 4 × 4 Convolution + BN + ReLu, Output: 4 × 4, Num_c: 512 |
| Concatenation: Input (4 × 4 × 512), (4 × 4 × 512), Output (4 × 4 × 1024) |
| Upsampling 4 × 4 Convolution + BN + ReLu, Output: 8 × 8, Num_c: 512 |
| Concatenation: Input (8×8×512), (8×8×512), Output (8 × 8 × 1024) |
| Upsampling: 4 × 4 Transpose Convolution + BN + ReLu, Output: 16 × 16, Num_c: 512 |
| Concatenation: Input (16 × 16 × 512), (16 × 16 × 512), Output (16 × 16 × 1024) |
| Upsampling: 4 × 4 Transpose Convolution + BN + ReLu, Output: 32 × 32, Num_c: 256 |
| Concatenation: Input (32 × 32 × 256), (32 × 32 × 256), Output (32 × 32 × 512) |
| Upsampling: 4 × 4 Transpose Convolution + BN + ReLu, Output: 64 × 64, Num_c: 128 |
| Concatenation: Input (64 × 64 × 128), (64 × 64 × 128), Output (64 × 64 × 256) |
| Upsampling: 4 × 4 Transpose Convolution + BN + ReLu, Output: 128 × 128, Num_c: 64 |
| Concatenation: Input (128 × 128 × 64), (128 × 128 × 64), Output (128 × 128 × 128) |
| Upsampling: 4 × 4 Transpose Convolution, Output: 256×256, Num_c: 3 |

These networks require all the input information to pass through each of the middle layers. In most of the image-to-image translation problems, it is desirable to share the feature maps across the network since both input and output images represent the same underlying structure. For this purpose, we added a skip connection while following the general configuration of a "U-Net" [44]. Skip connections simply concatenate the channels at the ith layer with the channels at the (n–i)th layer.

- Discriminator Network

We adapted PatchGAN architecture [45] for the discriminator, which penalized the structure at the scale of patches. It tried to classify each N × N patch as either real or fake. Final output of the discriminator (*D*) was calculated by averaging the received responses by running the discriminator convolutionally across the image. In this case, the patch was 30 × 30 in size, and each convolutional layer was followed by a ReLU activation and batch normalization. Zero-padding layers were used to preserve the edge details of the input feature maps during the convolution. Discriminator architecture is described in Table 2.

**Table 2.** Discriminator architecture of the CGANet model.

| Discriminator Architecture |
| :---: |
| Input Image (256 × 256 × 3) + Target Image (256 × 256 × 3) |
| Concatenation: Input (256 × 256 × 3), (256 × 256 × 3), Output (2 × 2 × 1024) |
| Downsample 4 × 4 Convolution + BN + ReLu, Output: 128 × 128, Num_c: 64 |
| Downsample 4 × 4 Convolution + BN + ReLu, Output: 64 × 64, Num_c: 128 |
| Downsample 4 × 4 Convolution + BN + ReLu, Output: 32 × 32, Num_c: 256 |
| Zero Padding 2D: Output: 34 × 34, Num_c: 256 |
| Downsample 4 × 4 Convolution + BN + ReLu, Output: 31 × 31, Num_c: 512 |
| Zero Padding 2D: Output: 33×33, Num_c: 512 |
| Downsample 4 × 4 Convolution + BN + ReLu, Output: 30 × 30, Num_c: 1 |

## 4. Experimental Details

This section discusses the experimental details of our proposed CGANet model and the quality matrices used to evaluate the performance of the proposed model. CGANet performance is compared with two other state-of-the-art methods: the Gaussian mixture model [18] and lightweight pyramid networks [17]. The algorithm implementation was conducted using Python and TensorFlow 2.0 [46]. CGANet was trained on a computer with a 2.2 GHz, 6-core Intel core i7 processor, 16 GB memory, and an AMD Radeon Pro 555X GPU.

### 4.1. Dataset

The training set consisting of 1500 images was chosen from a global road damage detection challenge dataset [47]. Rain streaks of different angles and intensities have been added to those images using Photoshop to create a synthesized rainy image set. Corresponding clean images become the target ground truth image set for the synthesized rainy image set. The test set consists of both synthesized and real-world rainy images. Three hundred synthesized images were chosen from the global road damage detection challenge dataset and pre-processed similarly when preparing the training set. Test dataset outputs are shown in Figure 2 as a comparison between the proposed CGANet model and the state-of-the-art de-raining methods. Real-world rainy images were taken from the internet, and they were considered only for demonstrating the effectiveness of the CGANet model. Since ground truth images were not available for the real-world rainy images, they were not taken into the account when training the model. Test results of real-world images are shown in Figure 3.

### 4.2. Evaluation Matrix and Results

The peak signal-to-noise ratio (PSNR) and structural similarity index (SSIM) [48] were used to evaluate and compare the performance of the model. PSNR measures how far the de-rained image is distorted from its real ground truth image by using the mean squared error at the pixel level. As shown in Table 1, the proposed CGANet model obtained the best PSNR value compared to the other two methods. The structural similarity index (SSIM) is a perception-based index that evaluates image degradation as the perceived difference in structural information while also incorporating both luminance masking and contrast masking terms. Table 3 shows the SSIM value comparison between the proposed CGANet model and the other two state-of-the art methods. By referring to this comparison, we could verify that the proposed method performed well compared to other de-raining mechanisms, and this is also visually verifiable in Figures 2 and 3.

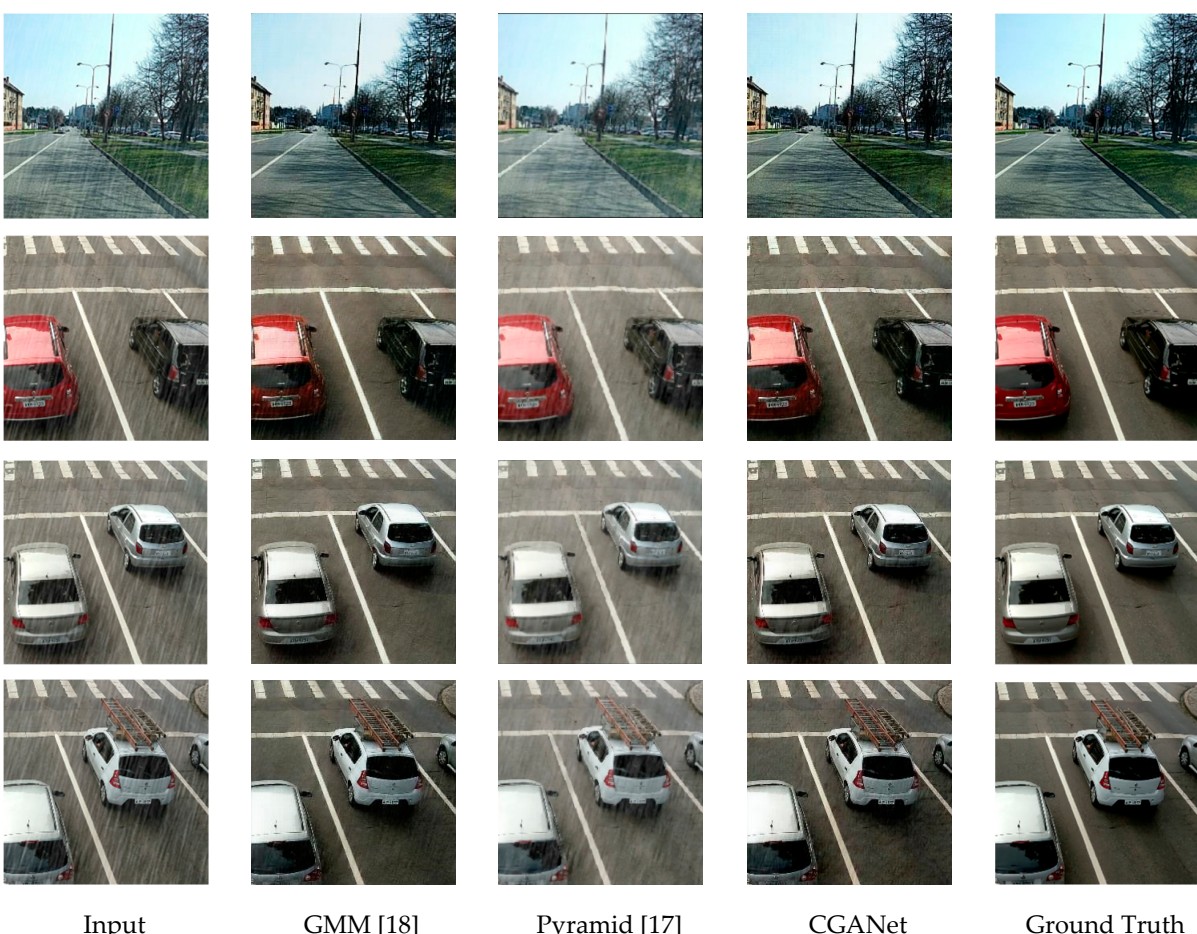

Input          GMM [18]          Pyramid [17]          CGANet          Ground Truth

**Figure 2.** Qualitative comparison between GMM, pyramid networks, and proposed CGANet methods.

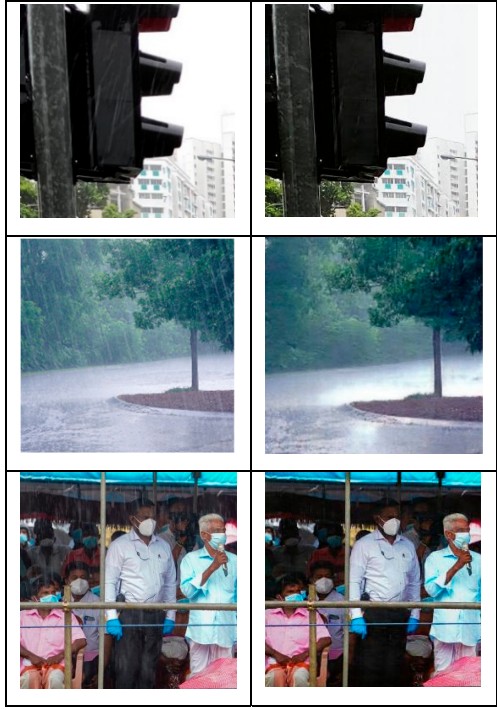

**Figure 3.** CGANet on real-world dataset ((**Left**) Input image; (**Right**) de-rained output).

**Table 3.** Quantitative comparison between different de-raining methods (mean $\pm$ STD).

| Index | Pyramid | GMM | CGANet |
|---|---|---|---|
| PSNR | $23.48 \pm 2.09$ | $24.37 \pm 2.15$ | $25.85 \pm 1.57$ |
| SSIM | $0.731 \pm 0.06$ | $0.762 \pm 0.06$ | $0.768 \pm 0.04$ |

*4.3. Parameter Settings*

To optimize the proposed model, we followed the findings provided in the original GAN paper [14]. Instead of training the generator to minimize $log(1 - D(x; G(x; z))$, we trained it to maximize $log\, D(x; G(x; z))$. Since the discriminator could be trained much faster compared to the generator, we divided the discriminator loss by 2 while optimizing the discriminator. As such, the discriminator training speed slowed down compared to the generator. Both the discriminator and generator models were trained with an Adam optimizer [49] with a learning rate of 0.0002 and a momentum parameter $\beta 1$ of 0.5 [15]. The model was trained using 150 epochs and updated after each image, and as such, the batch size was 1.

**5. Conclusions**

In this paper, we have proposed a single image de-raining model based on conditional generative adversarial networks and a Pix2Pix framework. The model consists of two neural networks: a generator network to map rainy images to de-rained images, and a discriminator network to classify real and generated de-rained images. Different performance matrices were used to evaluate the performance of the new model using both synthesized and real-world image data. The evaluations proved that the proposed CGANet model outperformed the state-of-the-art methods for image de-raining. The new CGANet model is presented as a high-potential approach for successful de-raining of images.

This paper is focused on image de-raining; however, the proposed model applies equally well to any other image translation problem in a different domain. In future developments, further analysis can be carried out to optimize the loss function by incorporating more comprehensive components with local and global perceptual information.

**Author Contributions:** Conceptualization, P.H., D.A., and R.N.; methodology, P.H., and R.N.; investigation, P.H.; data curation, P.H.; writing—original draft preparation, P.H.; writing—review and editing, R.N., D.A., D.D.S., and N.C.; supervision, D.A., D.D.S., and N.C. All authors have read and agreed to the published version of the manuscript.

**Funding:** This research received no external funding.

**Institutional Review Board Statement:** Not applicable.

**Informed Consent Statement:** Not applicable.

**Data Availability Statement:** Data and source code for the experiments are publicly available at https://github.com/prasadmaduranga/CGANet (accessed on 11 December 2020).

**Acknowledgments:** This work was supported by a La Trobe University Postgraduate Research Scholarship.

**Conflicts of Interest:** The authors declare no conflict of interest.

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
