# Peer review of "Rain Streak Removal for Single Images Using Conditional Generative Adversarial Networks"

_applsci, doi:10.3390/app11052214_

Round 1

Reviewer 1 Report

The manuscript proposes a cGAN based method for removal fo rain streak. Though the proposed method is interesting, the manuscript is incomplete 
in many aspects and needs significant improvement.

Here are more detailed comments:

1) The claimed contributions of the manuscript are overstated. Points 1-4 are more or less rephrasing of same contribution. Especially, talking about the
claim 3, patchGAN classifier has been used in such scenario in many previous works. I am not sure what does the authors mean by introducing? To summarize, please#
be careful about what you claim to be novel contribution.

2) The related works w.r.t GAN is not adequate and needs to be further augmented by discussing applications of GAN in differen fields. I would
recommend following works:
i) GAN-Based Rain Noise Removal From Single-Image Considering Rain Composite Models
ii) Unsupervised multiple-change detection in VHR multisensor images via deep-learning based adaptation
iii) Reliability of GAN Generated Data to Train and Validate Perception Systems for Autonomous Vehicles
iv) TriGAN: Image-to-Image Translation for Multi-Source Domain Adaptation
v) Ultrasound Image Classification using ACGAN with Small Training Dataset
vi) Semantic-Fusion Gans for Semi-Supervised Satellite Image Classification

Moreover some of the works mentioned above are related to the proposed method and clear difference w.r.t. them needs to be mentioned.

4) Current works related to the image common corruption are also relevant to this work and needs to be briefly discussed, e.g.,
Certified Adversarial Robustness via Randomized Smoothing

3) Please revise Figure 1 to remove the red underlines.

4) Equation 1-3 - while separating different entities generally , is used here . has been used which looks strange.

5) Line 169-194 is mere rephrasing of cGAN description found in many other previous works. Please expand it, explain in terms of rain streak removal why is it
special.

6) Introduction claims "A key limitation in the existing state of the art methods is that 42 they are designed to deal only with certain types of rain streaks" - however
I do not observe anything in the proposed method that can overcome this limitation. Please expand on this point.

7) "Real-world rainy images are taken from Google and they were considered only for demonstrating the effectiveness of CGANet model." will they be made publicly available?

8) Conclusion section needs to be improved by incorporating more thought-provoking discussion and indicating directions towards future works. 

Reviewer 2 Report

The paper presents the results of applied research in the field of AI R&D (Artificial Intelligence Research & Development). The research is based on the Generative Adversarial Networks (GAN) framework, more precisely, on the Conditional GAN (CGAN) modification, which allows accounting for additional information, which influences the process of output image generation. Authors develop a method of image de-raining, i.e., rain streak removal.
The work contains a good presentation of the problem at hand. The survey contains a wide range of methods applied to the problem, their classification, and a large number of references. The CGAN framework is also described in detail. The composition and volume of the generated training and test datasets appear to be quite reasonable.
The presented result is interesting, the presentation has no major issues, so the paper can be accepted for publication.
Minor issues:
1. every acronym and symbol requires definition (e.g., CGANNet, D(x.y), G(z,y), Ex.y.z,  âƒ¦… ⃦, Lambda(?))
2. the notation should be used correctly: e.g., in «latent variable ? defined on the latent space ?», different symbols should be used for the variable and the space of its values;
3. (2) and (3) requires correction (e.g., “L1”, “???? (?. ?)”, compare with (1))
4. (4) lacks parentheses;
5. It is advisable to add the algorithm implementation details to section 4 or, maybe, 3 (a brief description of the software implementation and the platform used for the calculations);
6. Despite the correct citation of the sources on the GAN-CGAN framework, it is necessary to clarify the format of the study performed, indicating in the annotation (briefly), in section 1 (introduction) (in detail) and section 5 (Conclusion) (also briefly) the sources of the parts of the algorithm and the specific author's contribution to the solution. The fact that the algorithm is based on some known instruments and applied to a new problem in no way diminishes the contribution and the interest to the article. Nevertheless, those known instruments must be presented correctly.

Round 2

Reviewer 1 Report

The revised manuscript is vastly improved. However I still have some concerns:

1) Contribution 4 needs to be removed. Evaluating is an integral part of a manuscript and is not counted as a contribution.

2) Line 277-279: please provide GPU details

3) Figure 2, when we look at the 2nd 3rd and 4th row:  in 2nd row CGANet loses some part of the white zebra crossing  while for similar input it retains white pixels 
accurately in the 3rd and 4th column. Please explain this anomaly?

4) Figure 3: the benefit of CGANet in real world dataset seems to be negligible!

5) Future work talks about incorporating more domain specific input - what is domain in this context?
